# *Bifidobacterium breve* Alleviates DSS-Induced Colitis in Mice by Maintaining the Mucosal and Epithelial Barriers and Modulating Gut Microbes

**DOI:** 10.3390/nu14183671

**Published:** 2022-09-06

**Authors:** Meng-Meng Niu, Huan-Xin Guo, Jun-Wu Cai, Xiang-Chen Meng

**Affiliations:** 1Key Laboratory of Dairy Science, Ministry of Education, Northeast Agricultural University, Harbin 150030, China; 2Food College, Northeast Agricultural University, Harbin 150030, China

**Keywords:** colitis, *Bifidobacterium breve*, exopolysaccharide, intestinal barrier, gut microbiota

## Abstract

This study was designed to explore the different intestinal barrier repair mechanisms of *Bifidobacterium breve* (*B. breve*) H4-2 and H9-3 with different exopolysaccharide (EPS) production in mice with colitis. The lipopolysaccharide (LPS)-induced IEC-6 cell inflammation model and dextran sulphate sodium (DSS)-induced mice colitis model were used. Histopathological changes, epithelial barrier integrity, short-chain fatty acid (SCFA) content, cytokine levels, NF-κB expression level, and intestinal flora were analyzed to evaluate the role of *B. breve* in alleviating colitis. Cell experiments indicated that both *B. breve* strains could regulate cytokine levels. In vivo experiments confirmed that oral administration of *B. breve* H4-2 and *B. breve* H9-3 significantly increased the expression of mucin, occludin, claudin-1, ZO-1, decreased the levels of IL-6, TNF-α, IL-1β and increased IL-10. Both strains of *B. breve* also inhibited the expression of the NF-κB signaling pathway. Moreover, *B. breve* H4-2 and H9-3 intervention significantly increased the levels of SCFAs, reduced the abundance of *Proteobacteria* and *Bacteroidea*, and increased the abundance of *Muribaculaceae*. These results demonstrate that EPS-producing *B. breve* strains H4-2 and H9-3 can regulate the physical, immune, and microbial barrier to repair the intestinal damage caused by DSS in mice. Of the two strains, H4-2 had a higher EPS output and was more effective at repair than H9-3. These results will provide insights useful for clinical applications and the development of probiotic products for the treatment of colitis.

## 1. Introduction

Inflammatory bowel disease (IBD) is an idiopathic intestinal inflammatory disease [1] that includes Crohn’s disease (CD) and ulcerative colitis (UC) [2]. IBD enteritis has a high morbidity, affecting nearly 6.8 million people [3,4]. It is generally believed that intestinal barrier dysfunction, gut microbiota changes, pathogen invasion, oxidative stress, immune abnormalities, and levels of inflammatory mediators can all be factors in the occurrence and development of IBD [5,6,7]. Emerging sequencing technologies and the application of innovative bioinformatics have recently brought attention to the role of the gut microbiome [8,9,10]. Accumulating evidence suggests that the richness of intestinal flora in IBD patients is reduced, the content of *Lactobacillus* and *Bifidobacterium* is reduced, and the intestinal *Escherichia coli* and *Clostridium* are increased [8,11,12,13]. Studies also confirmed that damage of the intestinal mucosal barrier is another factor that promotes the occurrence of IBD [14,15]. Dysregulation of the gut microbiota can drive changes in the proportion of mucin and O-glycome, which further disrupts the mucus layer, destroys host-microbe interactions, and ultimately leads to IBD pathogenesis [14,15]. Tight junctions (TJ) are important complexes between intestinal epithelial cells and play an important role in maintaining cell permeability and cell polarity [16]. Disruption of the mechanical barrier composed of TJ protein and adherens junction (AJ) proteins would activate pattern receptors of the intestinal immune barrier, and activation of the immune barrier is a manifestation of complete destruction of the intestinal barrier [17,18,19]. At present, the treatment method of IBD is mainly drug intervention. Despite some therapeutic benefits, this approach is not only expensive, but also has serious side effects. Therefore, there is currently a need to develop novel, low-side effect, and high-efficiency therapeutics for IBD treatment.

In previous studies, new treatments such as probiotics, prebiotics, and polyunsaturated fatty acids have effectively alleviated the occurrence of IBD with fewer side effects than conventional therapies [4,19,20]. Some probiotics have the same therapeutic effect as 5-aminosalicylic acid and are recommended by the European Society of Parenteral Enteral Nutrition (ESPEN) for the treatment of IBD [21]. With the deepening of clinical and animal experimental studies, bifidobacteria have been shown to alleviate intestinal inflammation by repairing the intestinal barrier, altering intestinal flora, and changing the levels of cytokines [19,22,23]. However, strain-specificities of bifidobacteria in alleviating intestinal inflammation, repairing the intestinal barrier and reducing adverse reactions were noted [24]. Strains with different characteristic metabolites can modulate intestinal inflammation in different ways. Conjugated linoleic acid-producing *B. breve* were shown to reduce intestinal mucosal injury and prevent further deterioration of DSS induced colitis in clinical studies [23]. The extracellular protein secreted by *Bifidobacterium longum* affected the composition of gut microbiota significantly as a regulatory factor of intestinal microbial communication, thereby maintaining gut microbial homeostasis [25].

Exopolysaccharide (EPS) is a long-chain high-molecular-mass polymer that is secreted outside the cell wall or on the cell surface during the growth of microorganisms [26]. Recently, EPS produced by bifidobacteria has attracted attention as a potential therapeutic agent due to its biological activities and safety [27,28]. EPS molecules can participate in the protection and niche colonization of bacteria and mediate the interaction between gut microbes and the host [29]. EPS can also affect intestinal inflammation by regulating intestinal flora, repairing the intestinal barrier, and promoting intestinal immunity [30,31].

However, studies found strain-specific effects—bifidobacteria strains differing in their ability to produce EPS had differences in intestinal flora regulation and intestinal barrier repair. For instance, compared with EPS deletion mutant *B. breve* UCC2003del, EPS-producing UCC2003 can more effectively change the structure of the intestinal flora and the ratio of short-chain fatty acid (SCFA) [32,33]. Ropy EPS-producing *Bifidobacterium longum* subsp. *longum* YS108R alleviates DSS-induced colitis by maintaining the mucosal barrier and modulating gut microbiota, but the non-ropy EPS producing strain C11A10B did not have this ability [34]. Previous experiments suggested that EPS secretion can affect the ability of different strains to repair intestinal damage and treat colitis in mice. However, the differential repair effects of different EPS-producing strains have not been thoroughly investigated. Therefore, this study compared the effects of two *B. breve* strains with different EPS yields both in vitro, using the LPS-induced IEC-6 cell inflammation model, and in vivo, using the DSS-induced colitis mouse model.

## 2. Materials and Methods

### 2.1. Preparation of Bacterial Strains and Cells

*B. breve* H4-2 and *B. breve* H9-3 with EPS production of 552.53 ± 12.54 and 363.17 ± 14.67 mg/L were isolated from the feces of exclusively breastfed infants and stored at the Key Laboratory of Dairy Science (KLDS), Ministry of Education, China. The strains were anaerobically incubated in De Man, Rogosa and Sharpe (MRS) (Oxiod, Basingstoke, UK) medium supplemented with 0.05% L-cysteine hydrochloride (mMRS) at 37 °C for 18 h and were cultured twice prior to the experiment. The extraction of EPS was performed using a method described previously with slight modifications [26]. Data are expressed as the mean ± SD of 3 biological replicates.

The human colon cell line Caco-2 cells were obtained from the Chinese Academy of Sciences (Shanghai, China) and the IEC-6 cells were purchased from the Zhejiang Mason Cell Technology Co., Ltd. (Jinhua City, Zhejiang Province, China). The Caco-2 cells and IEC-6 cells were cultured in the recommended DMEM medium with 10% fetal bovine serum, and 0.1 mg/mL penicillin/streptomycin. The cells were kept in a humidified 37 °C incubator with 5% CO_2_.

### 2.2. Analysis of Probiotic Characteristics of Strains

The Caco-2 adhesion assay was measured by the method described previously with some modifications [35]. In brief, Caco-2 cells were seeded in 12-well culture plates at the concentration of 2 × 10^5^ cells per well. Bacteria were seeded at a concentration of 10^8^ CFU/well. Each well was washed four times with PBS after co-culture for 2 h, then trypsin (Gibco, New York, NY, USA) was used to separate the cells. After a 10-fold serial dilution, bound bacteria were spread on mMRS agar plates. The adhesion rate was evaluated based on the number of colonies grown on MRS agar using the following formula:$$\mathrm{Adhesion}\;\mathrm{rate}\;(\%)=\frac{C1}{C0}\;\times \;100$$

C1 represents the total count of adhesive cells after treatment, and C0 represents the total count of cells before treatment.

The survival rate of strains in simulated gastrointestinal digestion was evaluated using the modified method previously determined [34]. Briefly, bacterial cells (1.0 × 10^9^ CFU/mL) were suspended in saliva for 5 min, then centrifuged at 8000 rpm for 5 min, and then resuspended in gastric fluid and incubated for 2 h. Finally, the bacterial cells were centrifuged again, resuspended in intestinal fluid and incubated for 2 h. The viable cells were determined by plate counting using the following formula:$$\mathrm{Survival}\;\mathrm{rate}\;(\%)=\frac{\mathrm{LogN}1}{\mathrm{LogN}0}\times 100$$
where N1 represents the total count of strains after treatment, and N0 represents the total count of strains before treatment.

### 2.3. Assays of the Cell Proliferation and Cytokine Level Test

IEC-6 cells (about 1.25 × 10^5^ cells/mL) were seeded into 96-well plates and incubated for 12 h, and then the cells were co-cultured for 24 h with 10:1, 100:1, and 1000:1 bacterial cells:IEC-6 cells. IEC-6 cells were also co-cultured for 24 h with 20 μg/mL LPS and bacterial cells (because this was found to be the best proliferation effect ratio) to test cell viability under LPS stimulation. After the incubation, 20 μL CCK-8 solution (2 μL CCK-8 in 18 μL medium) was added to each well, and IEC-6 cells and bacterial cells were incubated for another 1 h. Then, the microplate reader was used to measure OD_450_ values (Bio-Rad Laboratories, Hercules, CA, USA). Viability value (%) was calculated as [(b − c)/(a − c)] × 100, where a and b are respective OD_450_ values without and with bifidobacterial cells treatment, while c is OD_450_ value of the test reagent. The control cells with medium treatment only were set at 100% viability.

IEC-6 cells (about 1.25 × 10^5^ cells/mL) were co-cultured for 24 h with 20 μg/mL LPS and 1 mL bacteria (best proliferation effect ratio). The supernatant was collected to detect levels of cytokines (IL-8 and IL-10), which were determined using a commercial ELISA kit according to the manufacturer’s instructions (Nanjing Jiancheng Biotechnology Institute, Nanjing, China).

### 2.4. Animals and Treatments

All animal experiments were strictly carried out according to the Laboratory Animal-Standards of Welfare and Ethics and were approved by the Institutional Animal Care and Use Committee at Northeast Agricultural University. Forty male C57BL/6J mice (6-weeks old) were purchased from the Beijing Charles River Laboratory Animal Technology Co., Ltd. (Beijing, China), maintained under standard laboratory conditions: 25 ± 2 °C, 50 ± 10% humidity, and a 12 h/12 h light/dark cycle. Before conducting the experiments, animals were acclimatized to laboratory conditions for seven days. After 7 days of acclimatization to the environment, forty mice were equally divided into five groups. The experimental process was shown in Table 1. In brief, 2.5% DSS (molecular weight 36,000–50,000 Da, MP Biomedicals Ltd., Santa Ana, CA, USA) was added to drinking water to induce mice colitis, and 75 mg/kg/day of mesalazine was used as a positive control. Cells of 10^9^ cfu/day H4-2 and H9-3 were given to each mouse in H4-2 group and H9-3 group, respectively.

### 2.5. Inflammatory Markers and Histopathology

The DAI of each mouse was assessed on the last day of experiment to determine hematochezia, stool consistency, and weight loss (Table 2).

### 2.6. Determination of D-Lactic Acid in Serum

The concentrations of D-lactic acid in serum were measured with the ELISA kit (Beijing Chenglin Bioengineering Institute, Beijing, China) according to the manufacturer’s instructions.

### 2.7. Assessment of Myeloperoxidase (MPO) Activity

The changes of MPO activity in colon tissue were analyzed using a Myeloperoxidase Test Kit according to the manufacturer’s instructions (Nanjing Construction Co., Ltd., Nanjing, China).

### 2.8. Evaluation of the Histological Score

Fresh colon tissues were fixed in 4% paraformaldehyde for 24 h, decolorized with ethanol, and embedded in paraffin. Then, embedded samples were cut into 4 um sections, dewaxed and stained with hematoxylin-eosin (H&E) and alcian blue and periodic acid-schiff (AB-PAS), and finally observed the histopathological changes of different groups of colons under a 200× microscope (OLYMPUS BX53, OLYMUPS, Tokyo, Japan) [36]. The pathological evaluation was based on Table 3; density of goblet cells was calculated by Image-pro plus 6.0 (Media Cybernetics, Inc., Rockville, MD, USA).

### 2.9. Determination of Biochemical Indices in Serum and Colon

The levels of catalase (CAT), glutathione (GSH), superoxide dismutase (SOD) and malondialdehyde (MDA) in serum were evaluated via commercial kits according to the manufacturer’s instructions (Nanjing Construction Co., Ltd., Nanjing, China). The levels of IL-6, IL-8, IL-10, TNF-α and IL-1β in colon tissues were evaluated by ELISA assay kits (Beijing Chenglin Bioengineering Institute, Beijing, China) according to the manufacturer’s instructions.

### 2.10. Immunohistochemical Staining and Scoring

Immunohistochemistry (IHC) was used to detect the expression of ZO-1, occludin, and claudin-1 in colonic tissues according to the method of Wang et al. with minor modification [37]. Briefly, the embedded paraffin cuts were sliced into 4 μm sections with a microtome, then dewaxed and hydrated, placed in a mixed antigen repair solution, and then incubated with 3% hydrogen peroxide to block the endogenous peroxidase. After incubation in 1% BSA blocking solution for 30 min, the sections were incubated overnight with primary antibodies ZO-1 (1:400, Abcam, Cambridge, UK), occludin (1:100, Abcam) and claudin-1 (1:400, Abcam), and then incubated 1 h with a secondary antibody (Millipore, Billerica, MA, USA). Subsequently, the reaction was terminated with 2% PBST after color development with DAB staining. Finally, the positive area of DAB staining was observed under a microscope after dehydration treatment with hematoxylin staining.

The IHC scores of TJ proteins ZO-1, occludin, and claudin-1 were measured using IHC profiles in the ImageJ program. The IHC scores were assigned as follows: 4 for the high positive zone, 3 for the positive zone, 2 for the low positive zone, and 1 for the negative zone [38].

### 2.11. Quantitative Real-Time PCR

Total colon tissues RNA was extracted by the Total RNA Kit (Vazyme, Nanjing, China). Reverse transcription was performed using the GoScript™ Reverse Transcription Mix kit (Promega Biotech Co., Ltd., Beijing, China). The PCR reactions were performed using GoTaq^®^ qPCR Master Mix kit (Promega Biotech Co., Ltd., Beijing, China) on the PikoReal 96 Real-Time PCR system (Thermo Scientific, Waltham, MA, USA). The MUC1, MUC2, ZO-1, occludin, claudin-1, TLR2, TLR4, TLR6, NF-κB and IκBα relative mRNA expressions were calculated according to the 2^−ΔΔCt^ method, comparing with the expression of GAPDH housekeeping gene. Primers are listed in Table 3.

### 2.12. SCFAs Quantification

SCFAs levels of colon contents were measured according to previous studies with minor modifications. Colon contents (0.1 g) were diluted with 0.5 mL deionized water, then 200 μL 50% sulfuric acid, 50 μL 500 mg/L internal standard (cyclohexanone) solution, and 1 mL ether was added. The mixture was homogenized for 1 min and centrifuged at 12,000 rpm for 20 min to extract the supernatant. The supernatant was analyzed using gas chromatography-mass spectrometry (Shimadzu GCMS-QP2010 Ultra system). The chromatographic system was an Agilent DB-WAX capillary column (30 m × 0.25 mm × 0.25 μm) with high-purity helium as the carrier gas at a flow rate of 1.0 mL/min. The injection port temperature was 220 °C, and the injection volume was 1 μL. The mass spectrometry system was an electron bombardment ion source (EI) with an ion source temperature of 230 °C and an interface temperature of 220 °C.

### 2.13. Gut Microbiota Analysis

According to the manufacturer’s instructions, the total bacterial DNA of the colon contents was extracted using the Fast DNA SPIN extraction kit (MP Biomedicals, Santa Ana, CA, USA). Amplification of the region encompassing the bacterial 16S rRNA genes V3-V4 was achieved using the forward primer 338F (5′-ACTCCTACGGGAGGCAGCA-3′) and the reverse primer 806R (5′-GGACTACHVGGGTWTCTAAT-3′), and then Agencourt AMPure Beads (Beckman Coulter, Indianapolis, IN, USA) and PicoGreen dsDNA Assay Kit (Invitrogen, Carlsbad, CA, USA) were used to purify and quantify PCR amplicons. After the individual quantification step, amplicons were pooled in equal amounts, and pair-end 2 × 300 bp sequencing was performed using the Illumina MiSeq platform with the MiSeq Reagent Kit v3 at Shanghai Personal Biotechnology Co., Ltd. (Shanghai, China). The Quantitative Insights Into Microbial Ecology (QIIME, v1.8.0) pipeline was employed to process the sequencing data, as previously described [39].

### 2.14. Statistical Analysis

All values are expressed as the mean ± standard deviation (SD). The statistical significance of data comparisons was evaluated by one-way analysis of variance (ANOVA), followed by Duncan’s multiple range test using SPSS 20.0 statistical software. Values of *p* < 0.05 were considered to be statistically significant.

## 3. Results

### 3.1. Gastrointestinal Tolerance and Adhesion Characteristics of Strains

As shown in Table 4, three strains, including the reference strain BB12, had strong gastrointestinal tolerance and Caco-2 adhesion properties. H4-2 exhibited higher gastrointestinal tolerance (71.05%) and adhesion (48.66%) than BB12. H9-3 showed no significant difference from BB12 in gastrointestinal tolerance and adhesion (*p* > 0.05).

### 3.2. Intervention with B. breve Reduced Inflammation in LPS Induced IEC-6 Cells

The in vitro anti-inflammatory effects of the two strains were preliminarily evaluated using the IEC-6 cell inflammation model. When the number of viable bacteria was 10^7^ cfu/mL (bacterial cells: IEC-6 cells = 100:1), both bifidobacterial strains effectively promoted the growth of IEC-6 cells (Figure 1A), and the proliferation effect was good among all concentrations tested. The 100:1 bacterial cell: IEC-6 cell ratio was adopted for subsequent in vitro experiments. When stimulated by 20 μg/mL LPS, cell viability decreased significantly (*p* < 0.05), and the intervening of *B. breve* strains H4-2 and H9-3 significantly increased cell viability (*p* < 0.05) (Figure 1B). Changes in cytokine levels are indicative of the degree of cellular inflammation. As shown in Figure 1C,D, treatment with LPS significantly increased the level of pro-inflammatory factor IL-8 and significantly decreased the level of anti-inflammatory factor IL-10. With the intervention of H4-2 and H9-3, pro-inflammatory cytokines decreased, and anti-inflammatory cytokines increased. Compared with the LPS group, the group that received H4-2 had significantly lower IL-8 content and significantly higher IL-10 content (Figure 1C,D) (*p* < 0.05). There was no significant difference in IL-8 and IL-10 levels between the LPS group and the group that was treated with H9-3 (*p* > 0.05). Together, these results indicate that both H4-2 and H9-3 had potential inflammatory effects.

### 3.3. B. breve Improved Colitis Symptoms

The direct manifestation of IBD inflammation is weight loss. Compared with the control group, the weight of the mice in the DSS group decreased by nearly 25%. DSS also caused 12.5% mortality (Appendix A). The melasazine, H4-2 and H9-3 interventions reduced weight loss and mice mortality (Figure 2A), especially from days 3 to 5 after DSS intervention.

The colitis DAI, consisting of body weight loss, diarrhea, and bloody feces, was evaluated to further explore whether administering H4-2 and H9-3 alleviates the deterioration of IBD. The DAI of DSS group mice was significantly higher than that of the control group (*p* < 0.05) (Figure 2B). However, the melasazine, H4-2 and H9-3 intervention significantly decreased the DAI of the mice compared with DSS group (*p* < 0.05). Shortening of colon length can also be used as another indicator of worsening of IBD. Compared with the control group, the colon length of the mice in the DSS group was significantly shortened (*p* < 0.05) (Figure 2C), while intervention with melasazine, H4-2 and H9-3 significantly reduced shortening of the colon (*p* < 0.05).

The level of D-lactic acid is often used to evaluate changes in intestinal permeability. Compared with the control group, mice colons’ D-lactic acid concentration in DSS group had significantly increased (*p* < 0.05), while mesalazine, H4-2 and H9-3 interventions significantly decreased mice colons’ D-lactic acid concentration (*p* < 0.05) (Figure 2E). Similarly, MPO activity of DSS mice colon significantly increased compared with the control group (*p* < 0.05), while MPO activity significantly decreased under administration of mesalazine, H4-2 and H9-3 (*p* < 0.05) (Figure 2F). Notably, the effect of H4-2 was similar to the effect of mesalazine, indicating that H4-2 could effectively improve DSS-induced changes in intestinal permeability. Considering the effects on body weight, DAI, colon length, D-lactic level and MPO activity, it is clear that DSS significantly induced the occurrence of IBD. However, the intake of mesalazine, H4-2 and H9-3 can effectively alleviate the further deterioration of IBD.

### 3.4. B. breve Recovered the Damage in Colonic Tissue

The intestinal mucosa epithelial cells of the mice in the control group were intact and regular, with no necrosis or shedding and no infiltration of inflammatory cells. In contrast, DSS caused intestinal submucosal edema, loose epithelium, damaged crypts and inflammatory cell infiltration (Figure 3A). However, the intake of mesalazine, H4-2, and H9-3 each clearly reduced the disruption of colonic tissues integrity compared with the DSS group. Similarly, the histological scores of the DSS group were significantly increased compared with the control group (*p* < 0.05) (Figure 3B). The histological scores of the mesalazine and H4-2 and H9-3 groups were significantly decreased (*p* < 0.05).

### 3.5. B. breve Recovered Mucus Disruption and Goblet Cells Exhaustion

Intestinal goblet cells contents and mucin secretion levels were analyzed using AB-PAS staining to further verify the alleviating effect of strains on the intestinal barrier disruption. Mucins were abundant in goblet cells of the control group (Figure 4A), and they were mainly distributed on the surface of colonic epithelial cells. In contrast, in DSS group mice, the number of goblet cells was obviously decreased due to the damage to the inner and outer mucous layers of the colonic epithelium (Figure 4B). However, with the administration of mesalazine, H4-2 and H9-3, the goblet cell counts significantly improved (*p* < 0.05). Loss of the mucus layer components can trigger colitis. Therefore, the expression of MUC1 and MUC2 in the colon of mice after DSS intervention were analyzed (Figure 4C,D). Compared with the control group, the mice treated with DSS had significantly lower mRNA expression of MUC1 and MUC2 (*p* < 0.05). The administration of mesalazine, H4-2 and H9-3 significantly reduced this loss.

### 3.6. B. breve Improved Serum and Colon Biochemical Functions

Oxidative stress is another trigger for inflammation. The levels of CAT, GSH, MDA and SOD were measured in serum to further evaluate the degree of oxidative stress caused by DSS in mice. Compared with the control group, the levels of CAT, GSH and SOD in the DSS group dramatically decreased (*p* < 0.05) (Figure 5A–C), while the level of MDA increased (*p* < 0.05) (Figure 5D). H4-2 and H9-3 administration significantly increased the levels of CAT and GSH relative to the DSS group (*p* < 0.05). In addition, H4-2 also could significantly decrease MDA content. H4-2 and H9-3 increased the SOD level, but the difference was not significant (*p* > 0.05).

Likewise, the content of cytokines in colonic tissue can also reflect the severity of inflammation or inflammatory cells infiltration caused by DSS. Compared with the colonic tissue of the control group, the colonic tissue of DSS group mice had higher IL-6, IL-1β, and TNF-α levels (*p* < 0.05) and lower IL-10 level (*p* < 0.05) (Figure 5E–I). These effects were significantly improved after administration of mesalazine, H4-2, and H9-3.

### 3.7. B. breve Protected the Epithelial Barrier

As shown in Figure 6A, IHC staining images of ZO-1, occludin and claudin-1 in the DSS group have almost no positive cells, and the IHC score significantly decreased compared with the control group (*p* < 0.05). The administration of mesalazine, H4-2 and H9-3 each significantly reduced this damage. The expression of ZO-1 mRNA in H4-2 and H9-3 intervention group was significantly higher than that in the mesalazine group (*p* < 0.05). Claudin-1 was mainly observed in the membranes of regenerated colonic epithelial cells, and its expression in the H4-2 group was similar to its expression in the control group, indicating that H4-2 repaired claudin-1 damage in mice colon tissue caused by DSS (Figure 6A). Similar results obtained from the IHC scores and qPCR expression further demonstrate the effect of H4-2 repair (Figure 6B,C). Together, the expression of three TJ proteins and mRNA, suggest that H4-2 and H9-3 repaired damage to the epithelial barrier caused by DSS, and the repair effect of H4-2 was significantly stronger than that of H9-3.

### 3.8. B. breve Regulated Toll-like Receptors (TLRs) and NF-κB Signaling Pathways

TLRs are important non-catalytic transmembrane proteins that can recognize characteristics composition of microorganisms and activate downstream signaling pathways, such as NF-κB [40]. Therefore, the expression of multiple TLRs and NF-κB inflammatory pathway-related genes in mouse colons were analyzed. Compared with the control group, the DSS group had significantly higher expression levels of TLR2, TLR4, and TLR6 genes (*p* < 0.05) (Figure 7A–C). Intervention with mesalazine, H4-2, and H9-3 prevented an increase in TLR2, TLR4 and TLR6 mRNA, and the expression level of TLR4 in the H4-2 intervention group mice was similar to that in the control group.

After acting on TLRs receptor, bifidobacteria further inhibit the NF-κB signaling pathway to alleviate the colonic injury caused by DSS. Compared with control group mice, the DSS group mice had significantly higher expression of NF-κB and significantly lower expression of IκBα (*p* < 0.05) (Figure 7D,E). After treatment with mesalazine, H4-2 and H9-3, mRNA expression of NF-κB was significantly reduced, and IκBα expression was significantly increased (*p* < 0.05).

### 3.9. B. breve Improved SCFAs Level

Compared with the colons of the control group, the colons of DSS group mice had significantly lower levels of acetic acid (*p* < 0.05) and also had lower levels of propionic acid, butyric acid, and valeric acid (Figure 8A–D). The levels of SCFAs in mice colons were significantly increased after administration of H4-2 and H9-3. The contents of acetic acid, propionic acid, butyric acid and valeric acid in the H4-2 group were higher than that in the H9-3 group. This indicated that intervention with H4-2 promoted the growth of the strains with SCFAs-producing ability.

### 3.10. B. breve Regulated Gut Microbiota

Intestinal microflora in intestinal contents were analyzed. The alpha-diversity in the DSS group was obviously reduced compared to the control group (Figure 9A,B). The observed species and the Shannon index of mice treated with H4-2 and H9-3 were clearly increased, and the difference was not significant (*p* > 0.05). The results of the principal coordinates analysis (PCoA) showed that H4-2 and H9-3 groups were markedly different from the DSS group, while the H4-2 and control groups were closer in distance (Figure 9C).

At the phylum level, the most abundant microbiota in the intestines of the four groups of mice included *Verrucomicrobiota*, *Firmicutes*, *Bacteroidota*, and *Proteobacteria* (Figure 9D). *Firmicutes* and *Bacteroidota* decreased significantly and *Proteobacteria* increased significantly after administration of DSS, and the supplementation with H4-2 and H9-3 reversed these changes. At the genus level (Figure 9E), the DSS group had a significantly lower abundance of *Turicibacter*, *Clostridia_Ucg-014*, *Muribaculaceae* and *Lactobacillus*, and a significantly higher abundance of *Escherichia Shigella* and *Bacteroides* compared with the control group. Interventions with H4-2 and H9-3 had obviously restored these changes, which indicated that both H4-2 and H9-3 can each significantly regulate the intestinal flora of mice.

### 3.11. The Key Microbiota Associated with Colitis

A comparative network was constructed to further identify the key microbiota relating the occurrence and alleviation of colitis (Figure 10A,B). Network comparison between the control group and H4-2 intervention group showed that microbiota with critical correlation included *Akkermansia*, *Muribaculaceae*, *Bacteorides*, *Clostriodia* and *Alistipes*. *Escherichia-Shigella*, *Bactereides*, *Clostriodia*, *Muribaculaceae* and *Alistipes* were the key microbiota in the comparative network between the H4-2 intervention group and the DSS group.

## 4. Discussion

The ability of bacteria to survive in adverse environments and adhere to surfaces are vital requirements for potential probiotics. Exopolysaccharides, a major bacterial metabolite, play an important role in bacterial cell adhesion and environmental tolerance. In the current study, both H4-2 and H9-3 could secrete EPS and had the ability to adhere to Caco-2 cells. These results indicated that H4-2 and H9-3 were good candidates for probiotics.

LPS is a component of the Gram-negative bacterial cell wall and can induce the production of cytokines, thereby destroying the integrity of the intestinal barrier [41]. Cell experiments showed that both H4-2 and H9-3 significantly improved the viability of IEC-6 cells stimulated by LPS (*p* < 0.05). Meanwhile, both H4-2 and H9-3 reduced the levels of IL-8 and increased the levels of IL-10. H4-2 significantly reduced the level of IL-8 (*p* < 0.05), which indicated that it might have anti-inflammatory effects.

The DSS-induced mice colitis model was used to study the intestinal barrier repair effect of two strains. DSS disrupts epithelial cell function in mice and causes colonic inflammation [42]. Our experiments confirmed that 2.5% DSS caused mice clinical symptom such as weight loss, hematochezia, diarrhea, colon shortening, increased mucosal edema, inflammatory cell infiltration, and decreased goblet cells in mice, consistent with findings by Liu et al. [43,44]. The occurrence of inflammation was accompanied by changes of intestinal permeability. DSS significantly reduced intestinal permeability (*p* < 0.05). Intervention with H4-2, H9-3 significantly decreased D-lactate content. Intervention with H4-2, H9-3 also decreased DAI and the histological score. The activity of the MPO enzyme is an indicator of neutrophil function and activation. MPO activity reflects the degree of neutrophil infiltration in colon tissue, and MPO can produce hypochlorous acid, which destroys various target substances, and triggers the inflammatory response [45]. The MPO activity of the DSS group was significantly higher than that of the control group, but this trend was significantly improved in the groups that were treated with H4-2 and H9-3 (*p* < 0.05).

Accumulated studies have confirmed that the mucus layer and the epithelium together constitute a physical barrier that prevents the invasion of harmful substances [46]. MUC2 is the predominant component of mucus, and is mainly secreted by goblet cells [47]. MUC2 and O-glycome together constitute the mucus layer of the intestinal barrier [15]. DSS significantly reduced the number of goblet cells and the mRNA expression of MUC2 in mice colons, and disrupted the balance of the mucus layer. Intervention with mesalazine, H4-2 and H9-3 significantly mitigated this destruction. The integrity of the epithelium monolayer is also critical for the appropriate functioning of the intestinal barrier. It effectively intercepts the invasion of antigens and prevents the activation of abnormal immune responses [48]. Tight junctions are a crucial factor in the control of the intestinal epithelial barrier, and are also crucial in normal physiological functions such as maintaining cell-selective permeability. Intramembrane protein ZO-1, the transmembrane protein claudin-1 and occludin are representative tight junctions [49]. In this study, H4-2 and H9-3 increased the expression of ZO-1, claudin-1, and occludin mRNA and protein, indicating that both H4-2 and H9-3 contributed significantly to the repair of intestinal injury. H4-2 had a stronger effect in improving intestinal injury than H9-3. The degree of improvement was comparable to that of *Bifidobacterium pseudocatenulatum*, *B. breve* M1 and *Lactobacillus helveticus* KLDS1.8701 [24,50,51].

Abnormal activation of immune cells in the gut leads to the overproduction of pro-inflammatory cytokines, resulting in intestinal inflammation [52]. The inhibition of pro-inflammatory cytokines and secretion of anti-inflammatory cytokines can serve as molecular targets for interventions that aim to improve IBD symptoms by regulating the immune responses. Therefore, this study examined the contents of IL-6, IL-8, IL-10, TNF-α, and IL-1β in mice colons after intervention with H4-2 and H9-3. TNF-α is a pro-inflammatory factor that can cooperate with other cytokines to induce the release of inflammatory mediators in vivo [51]. Excessive production of IL-1β enhances the permeability of endothelial and epithelial cells, which in turn aggravates the inflammation of the intestinal mucosa [51]. In the present study, the levels of pro-inflammatory cytokines (TNF-α, IL-1β, IL-8, IL-6) were higher and the concentration of the anti-inflammatory cytokine (IL-10) was lower in the DSS group than that in the control group. H4-2 and H9-3 intervention significantly reversed this trend. H4-2 significantly decreased the level of TNF-α to levels similar comparable with those measured in the control group.

TLRs served as pattern recognition receptors for immune cells to recognize microbial components in the gut [53,54,55]. TLR2, TLR4, and TLR6 play an important role in symbiosis and identification of pathogenic microorganisms in vivo [34]. TLR2 is located on the apical membrane surface of intestinal epithelial cells and plays a vital part in repairing intestinal epithelial cells [55,56]. TLR2 combines with TLR6 subunit to form a dimer that identifies the relevant components on the surface of pathogenic bacteria [57]. TLR4 is located on the surface of the basolateral membrane, and it activates the NF-κB inflammatory pathway, as does TLR2 [58,59]. Therefore, the expressions of TLR2, TLR4, and TLR6 can indicate the presence of inflammation. In our study, colons from mice in the H4-2 and H9-3 groups had significantly lower expression of TLR2, TLR4, and TLR6 genes compared with colons from the DSS group mice, indicating that H4-2 and H9-3 significantly reduce intestinal permeability and inflammation, which also consistent with the results of the D-lactic acid levels.

NF-κB is a classical inflammatory signaling pathway, which usually exists in a stable form in combination with IκBα. Stimulated by proinflammatory factors, the complex can be degraded by phosphorylation and proteolyse to promote translocation of NF-κB to the nucleus where it binds to target genes and plays a role in various biological processes [60]. Therefore, the levels of IκBα and NF-κB could prove the activation of the NF-κB signaling pathway. Our results confirmed that H4-2 and H9-3 led to a decrease in the mRNA expression of NF-κB and an increase in the mRNA expression of IκBα relative to the DSS group, which indicated that H4-2 and H9-3 could inhibit the NF-κB signaling pathway, thereby alleviating intestinal inflammation in mice.

Short chain fatty acids are produced by microbiota and enzymes in the gut and are involved in a variety of metabolic and immune processes [4]. They also play an important role in building the intestinal barrier, reducing inflammation, and the proliferation of intestinal epithelial cells [61]. Butyric acid is a short chain fatty acid that plays an essential role in recovery from IBD enteritis. It has been reported that the presence of butyrate can indicate the expression of pro-inflammatory factors and promote the expression of TJ proteins [62,63]. Other studies have shown that butyrate can reduce the level of NF-κB nuclear translocation in macrophages in distal tissue sections from patients [64]. Our results showed that H4-2 significantly increased butyric acid content, inhibited the release of pro-inflammatory factors, and inhibited the NF-kB signaling pathway. This was consistent with the trend observed in *Ruminococcus*, which is known to produce butyric acid [65]. Upon stimulation by acetate and propionate, GPR43 up-regulates the expression of regulatory T cell transcription factor (Foxp3) and promotes the proliferation of colonic regulatory T cells (Treg), thereby promoting the production of the anti-inflammatory cytokine IL-10 [66]. In this study, intervention with H4-2 significantly increased the content of acetic acid in the intestinal tract of mice, which can be explained by the increase in acetic acid producing bacteria (*Bifidobacterium*, *Lactobacillus*) in the gut microbiota of H4-2 group mice [67]. The intestinal tracts of H9-3 group mice had increased acetic acid and butyric acid levels, but the difference was not significant (*p* < 0.05).

Gut microbiota play an important role in the development and treatment of IBD [68,69]. Gut microbiota imbalance is associated with changes in the intestinal mucosa and mucosal inflammation. In the present study, the gut microbial diversity of H4-2 and H9-3 intervention mice was significantly higher than in mice treated with DSS. Changing of *Proteobacteria* abundance is considered as a marker of gut microbiota imbalance [70,71]. DSS caused increased abundance of *Proteobacteria*, which was consistent with previous studies [72,73]. Abundance of *Proteobacteria* was significantly decreased under the intervention of H4-2 and H9-3. *Firmicutes* is known to have anti-inflammatory effects [5]. DSS caused reduced *Firmicutes* abundance [74], and intervention with H4-2 and H9-3 significantly restored this reduction.

*Escherichia-Shigella* are Gram-negative bacteria that can damage the immune system and aggravate intestinal infection, which is usually relevant to the pathogenesis of IBD [75]. Increased *Escherichia-Shigella* and *Bacteroides* abundance in the DSS group is indicative of inflammation in mice. Intervention with H4-2 and H9-3 significantly reduced this effect. Meanwhile, comparative network analysis also demonstrated that *Escherichia-Shigella* and *Bacteroides* were key microbiota in inducing colitis. *Muribaculaceae* showed the ability to regulate immune cells and reduce the levels of pro-inflammatory factors [76]. DSS caused decreased *Muribaculaceae* abundance; nevertheless, this was increased in the H4-2 and H9-3 administration group. Our study suggests that *Muribaculaceae* species are promising microorganisms for the treatment and relief of colitis.

*Ruminiclostridium* promotes the development of the immune system by degrading polysaccharides to generate acetic acid and butyrate [77]. In this study, the abundance of *Ruminiclostridium* in the H4-2 group was higher than in H9-3 group, which was consistent with the yield of exopolysaccharides from H4-2 and H9-3. We also found that *Alistipes* played a critical role in alleviating colitis in our study (Figure 10). Notably, the relationship between *Alistipes* and host health remains controversial. However, recent articles indicate that *Alistipes* species produce succinic acid, an organic acid that helps maintain intestinal homeostasis by accelerating cell differentiation, promoting tight junction assembly, and increasing the expression of preglucagon in intestinal L cells [78]. In general, our results indicated that H4-2 and H9-3 intervention prevented the mice gut microbiota changes induced by DSS.

Based on these results, we concluded that EPS-producing *B. breve* can improve intestinal inflammation by repairing the intestinal barrier and regulating the contents of intestinal flora and metabolites. Similar results were previously found in EPS-producing *B. breve* UCC2003 and YS108R [33,34]. Studies have shown that probiotics and their multiple metabolites had a dose-effect relationship in relieving colitis [23,79,80], suggesting that EPS accumulation might enhance the anti-inflammatory activity of probiotics. The anti-inflammatory effects of H4-2 with higher EPS production were more prominent than those of H9-3 in the current study, which indicated that a dose-effect relationship of EPS was possible during colitis repair. Thus, EPS production may be a factor in alleviating colitis in *B. breve* H4-2 and *B. breve* H9-3. However, the role of other beneficial metabolites and the growth of specific microorganisms promoted by certain EPS has not been evaluated. Additional knockout technology, spectroscopy, and metabolomics technology will be required to further investigate this possibility.

## 5. Conclusions

In summary, intervention with EPS-producing *B. breve* strains H4-2 and H9-3 ameliorated DSS-induced colitis in mice. The effect of H4-2 was stronger than that of H9-3. There were multiple mechanisms involved in the alleviation of DSS-induced colitis by H4-2 and H9-3, including restoring the imbalanced microbial barrier by regulating the structure of microbial flora, increasing short chain fatty acid levels, repairing the physical barrier of the intestine by increasing MUC2 content and TJ protein concentration, repairing the immune barrier of mice intestines by reducing the release of pro-inflammatory factors (TNF-α, IL-1β, IL-6, IL-8), and inhibiting the NF-κB signaling pathway (Figure 11). This study will contribute to an improved understanding of the role of EPS-producing *B. breve* H4-2 and H9-3 in intestinal barrier damage repair. It will also lay a foundation for the repair of colitis by probiotic metabolites, which can also be used as indicators for screening and discovering probiotics in the future.

## Figures and Tables

**Figure 1 nutrients-14-03671-f001:**
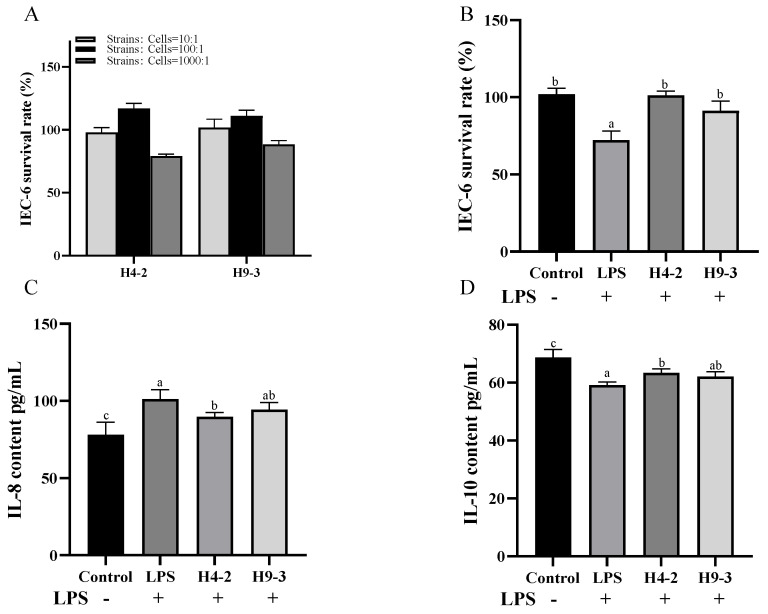
Viability of IEC-6 cells co-culturing with *B. breve* at different ratios of bacteria to cells (**A**) as well as survival rates (**B**), IL-8(**C**) and IL-10 (**D**) secretion of IEC-6 exposed to lipopolysaccharides (LPS) co-incubating with *B. breve*. Different lowercase letters (a, b, c) above the columns indicate significant differences (*p* < 0.05), and the same superscript letters indicate that the differences between the values are not statistically significant (*p* > 0.05).

**Figure 2 nutrients-14-03671-f002:**
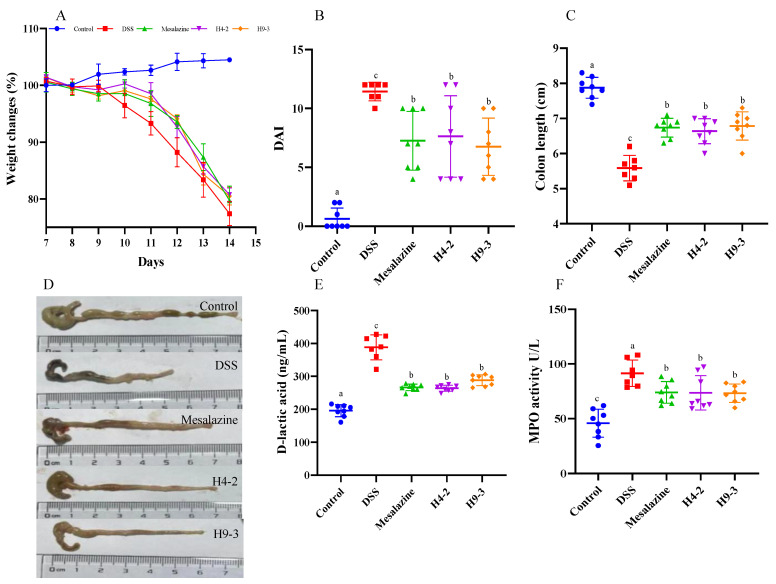
Changes of mice body weight (**A**), DAI (**B**), length of the colon (**C**), macroscopic pictures of colons (**D**), concentration of D-lactic acid (**E**) and activity of MPO enzyme in colons (**F**) after intervention with *B. breve*. Different lowercase letters (a, b, c) above the columns indicate significant differences (*p* < 0.05), and the same superscript letters indicate that the differences between the values are not statistically significant (*p* > 0.05).

**Figure 3 nutrients-14-03671-f003:**
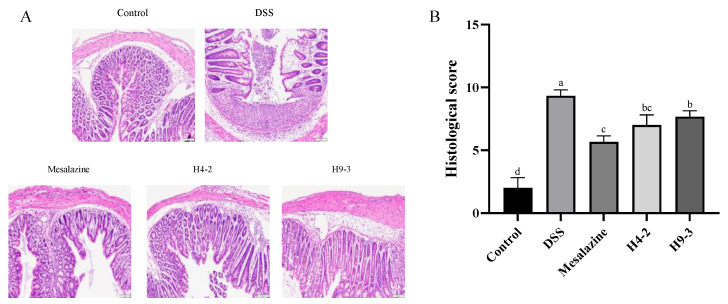
Effects of *B. breve* intervention on mice colon histological sections of H&E staining (**A**) and histopathology scores (**B**). Different lowercase letters (a, b, c, d) above the columns indicate significant differences (*p* < 0.05), and the same superscript letters indicate that the differences between the values are not statistically significant (*p* > 0.05).

**Figure 4 nutrients-14-03671-f004:**
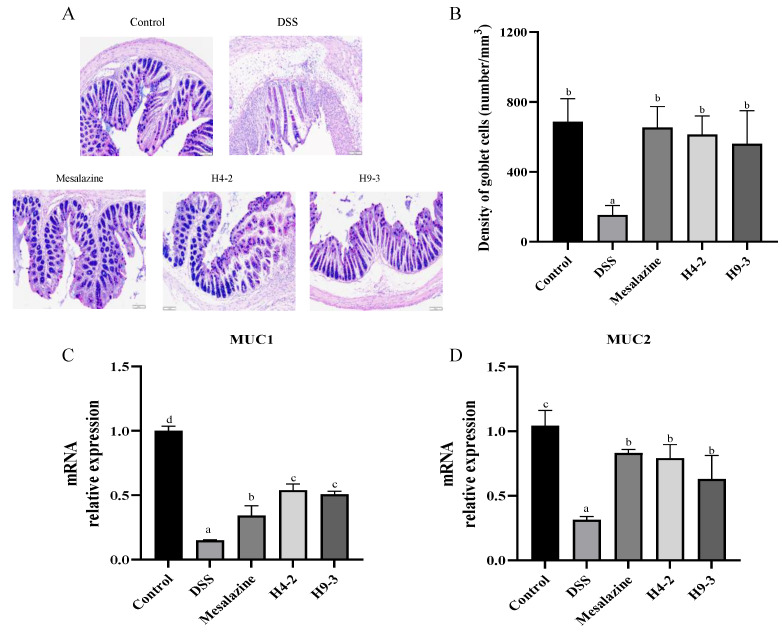
Effects of *B. breve* intervention on mice mucous layer including pathological section of AB-PAS staining (**A**), the number of goblet cells (**B**) and mRNA expression of MUC1 (**C**) and MUC2 (**D**). Different lowercase letters (a, b, c, d) above the columns indicate significant differences (*p* < 0.05), and the same superscript letters indicate that the differences between the values are not statistically significant (*p* > 0.05).

**Figure 5 nutrients-14-03671-f005:**
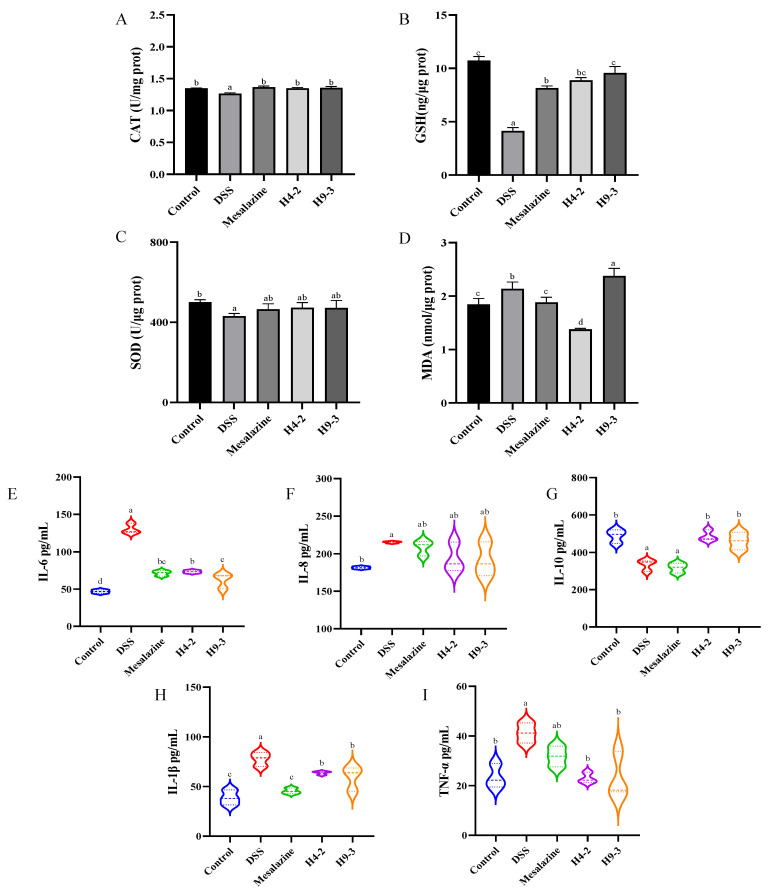
Changes of serum and colon biochemical functions in colitis mice after intervention with *B. breve*. Levels of CAT (**A**), GSH (**B**), SOD (**C**), and MDA (**D**) in serum; levels of IL-6 (**E**), IL-8 (**F**), IL-10 (**G**), TNF-α (**H**) and IL-1β (**I**). Different lowercase letters (a, b, c, d) above the columns indicate significant differences (*p* < 0.05), and the same superscript letters indicate that the differences between the values are not statistically significant (*p* > 0.05).

**Figure 6 nutrients-14-03671-f006:**
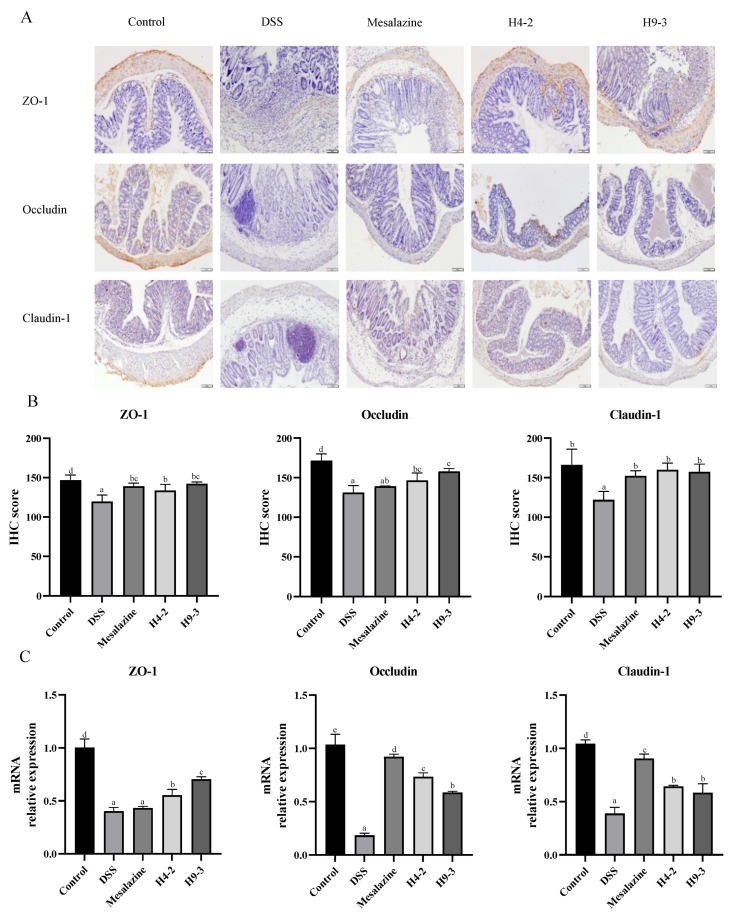
Effects of *B. breve* on the epithelial barrier. The IHC staining of TJ protein (**A**), IHC scores (**B**) and mRNA expression of TJ protein (**C**). Different lowercase letters (a, b, c, d) above the columns indicate significant differences (*p* < 0.05), and the same superscript letters indicate that the differences between the values are not statistically significant (*p* > 0.05).

**Figure 7 nutrients-14-03671-f007:**
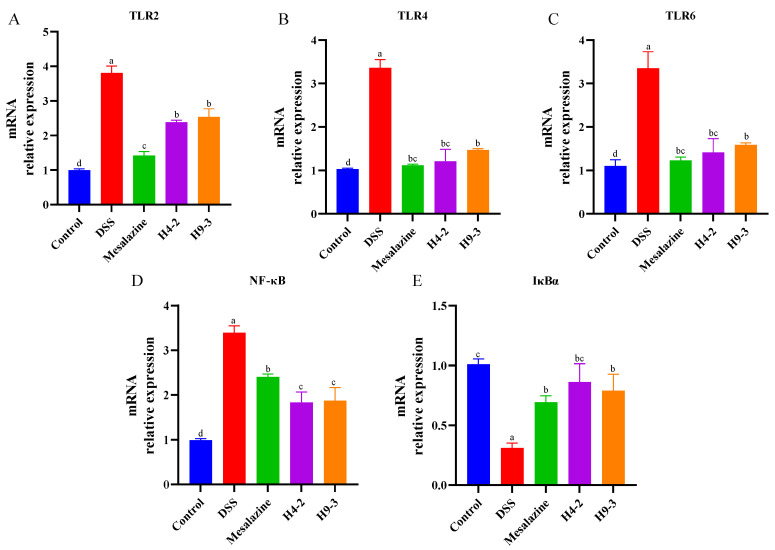
The mRNA expression changes of TLR-2 (**A**), TLR-4 (**B**), TLR-6 (**C**), NF-κB (**D**) and IκBα (**E**) in mice colon tissue after intervention with *B. breve*. Different lowercase letters (a, b, c, d) above the columns indicate significant differences (*p* < 0.05), and the same superscript letters indicate that the differences between the values are not statistically significant (*p* > 0.05).

**Figure 8 nutrients-14-03671-f008:**
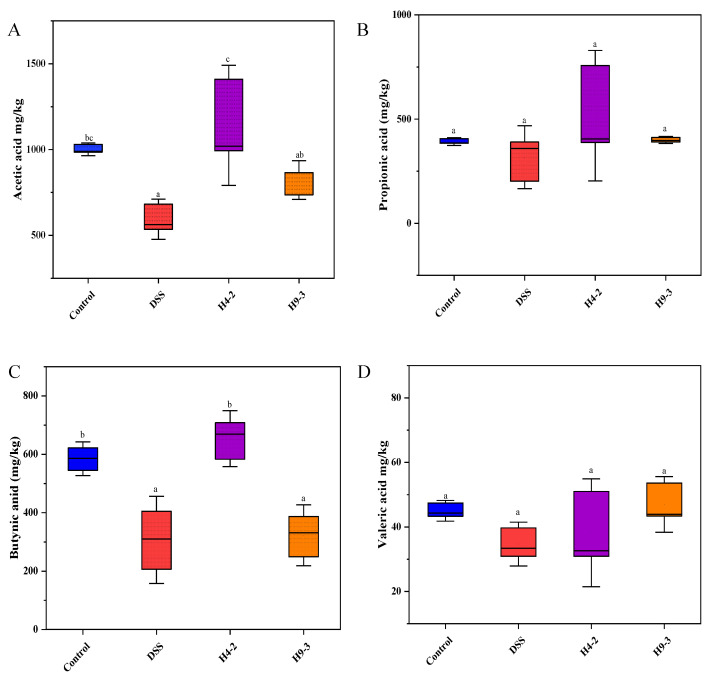
The concentrations of acetic acid (**A**), propionic acid (**B**), butyric acid (**C**) and valeric acid (**D**) in mouse intestine. Different lowercase letters above the columns indicate significant differences (*p* < 0.05), and the same superscript letters indicate that the differences between the values are not statistically significant (*p* > 0.05).

**Figure 9 nutrients-14-03671-f009:**
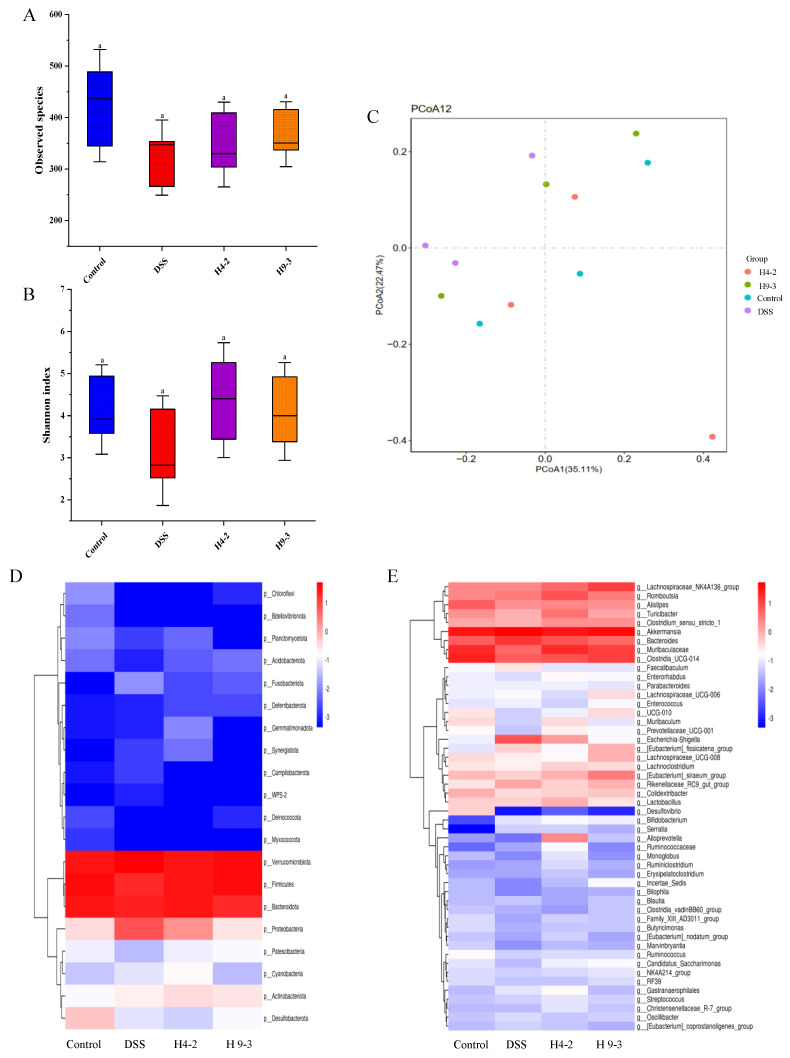
Changes of gut microbiota in mouse colon after intervention with *B. breve*. Comparison of the observed species (**A**), comparison of the Shannon index (**B**), PCoA analysis (**C**), phylum level (**D**), genus level (**E**). Different lowercase letters above the columns indicate significant differences (*p* < 0.05), and the same superscript letters indicate that the differences between the values are not statistically significant (*p* > 0.05).

**Figure 10 nutrients-14-03671-f010:**
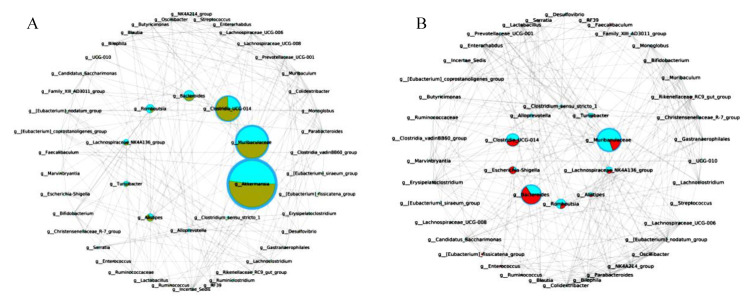
Correlation network analysis of microbial genus levels between *B. breve* H4-2 and the control (**A**). Correlation network analysis of microbial genus levels between *B. breve* H4-2 and DSS group (**B**).

**Figure 11 nutrients-14-03671-f011:**
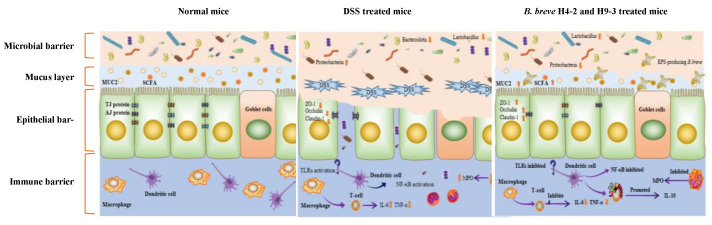
The schematic of improvement effects of EPS-producing *B. breve* on mice colitis.

**Table 1 nutrients-14-03671-t001:** Animal model experimental design.

Group	Daily Gavage Treatment (0.2 mL)	1–7 Days	8–14 Days
Control	Saline (0.9%)	Free drinking sterilized water	Free drinking sterilized water
DSS	Saline (0.9%)	Free drinking sterilized water	Free drinking 2.5% DSS water
Mesalazine	75 mg/kg/day mesalazine	Free drinking sterilized water	Free drinking 2.5% DSS water
H4-2	5 × 10^9^ cfu/mL H4-2	Free drinking sterilized water	Free drinking 2.5% DSS water
H9-3	5 × 10^9^ cfu/mL H9-3	Free drinking sterilized water	Free drinking 2.5% DSS water

**Table 2 nutrients-14-03671-t002:** Calculated disease activity index (DAI) score.

Score	Weight Loss (%)	Stool Consistency	Blood in Feces
0	none	normal	negative (no bleeding)
1	1.0–5.0	loose stools	negative
2	5.0–10.0	loose stools	hemoccult positive (slight)
3	10.0–15.0	diarrhea (slight)	hemoccult positive
4	15.0–20.0	diarrhea (watery diarrhea)	gross bleeding

**Table 3 nutrients-14-03671-t003:** Primer sequences used in real-time qPCR.

Genes	Primer Sequences (5′→3′)
GAPDH	Forward:5′-TCAAGAAGGTGGTGAAGCAG-3′ Reverse: 5′-AAGGTGGAAGAGTGGGAGTTG-3′
MUC1	Forward:5′-CATTCCAGACCACAATGGCTCCTC-3′ Reverse: 5′-ATTGACTTGGCACTGAAGGCTGAG-3′
MUC2	Forward:5′-TGGTCCAGGGTTTCTTACTCC-3′ Reverse: 5′-TGATGAGGTGGCAGACAGGAGAC-3′
ZO-1	Forward: 5′-CTTCTCTTGCTGGCCCTAAAC-3′ Reverse: 5′-TGGCTTCACTTGAGGTTTCTG-3′
Claudin-1	Forward: 5′- TCTACGAGGGACTGTGGATG-3′ Reverse: 5′-TCAGATTCAGCTAGGAGTCG-3′
Occludin	Forward: 5′- CACACTTGCTTGGGACAGAG-3′ Reverse: 5′-TAGCCATAGCCTCCATAGCC-3′
TLR-2	Forward: 5′-GACTCTTCACTTAAGCGAGTCT-3′ Reverse: 5′-AACCTGGCCAAGTTAGTATCTC-3′
TLR-4	Forward: 5′-GCCATCATTATGAGTGCCAATT-3′ Reverse: 5′-AGGGATAAGAACGCTGAGAATT-3′
TLR6	Forward: 5′-CAACTTAACGATAACTGAGAG-3′ Reverse: 5′-CCAGAGAGGACATATTCTTAG-3′
NF-κB	Forward: 5′-GCAAGCGTATCCCAAGAAGAGGTG-3′ Reverse: 5′-GCGATGCCTGCTACCACTCATTAC-3′
IκBα	Forward: 5′-TGGTGTGACTGTGGATCTCTGGAG-3′ Reverse: 5′-GGCTGGCTTCTCTGTGGTGATTC-3′

**Table 4 nutrients-14-03671-t004:** Gastrointestinal tolerance and adhesion characteristics of strains.

Strains	Survival Rate in Simulated Gastrointestinal Tract (%)	Adhesion Rate (%)
H4-2	71.05 ± 1.04 ^b^	48.66 ± 0.72 ^b^
H9-3	64.30 ± 0.96 ^a^	45.33 ± 0.17 ^a^
BB12	64.55 ± 0.50 ^a^	44.77± 1.56 ^a^

Different lowercase letters (a, b) above the columns indicate significant differences (*p* < 0.05), and the same superscript letters indicate that the differences between the values are not statistically significant (*p* > 0.05).

## Data Availability

The data presented in this study are available on request from the author.

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
