# Peer review of "Bifidobacterium breve Alleviates DSS-Induced Colitis in Mice by Maintaining the Mucosal and Epithelial Barriers and Modulating Gut Microbes"

_nutrients, 2022, doi:10.3390/nu14183671_

Round 1

Reviewer 1 Report

In the manuscript entitled “Alleviation effects of Bifidobacterium breve on DSS-induced colitis depending on mucosal and epithelial barrier maintenance and gut microbiota modulation”, submitted for review, I found almost the same title published earlier “Chen Y, Jin Y, Stanton C, et al. Alleviation effects of Bifidobacterium breve on DSS-induced colitis depends on intestinal tract barrier maintenance and gut microbiota modulation. Eur J Nutr. 2021;60(1):369-387. doi:10.1007/s00394-020-02252-x”. although authors in the present manuscript have taken different strains of Bifidobacterium breve to prove their hypothesis and presented lots of data. However, I feel the work is similar to previously published.

I have several comments that need to be answered

1.     Did the authors notice any mortality? Must be explained in the manuscript.

2.     Authors have done reduced glutathione, I will suggest if authors can add any glutathione-dependent enzymes such as GPx, GST, or GR…can improve the quality of the manuscript.

3.     I will suggest adding a graphical abstract, that can be a point of attraction for the readers.

Reviewer 2 Report

The authors describe the anti-inflammatory effects of the probiotic strains Bifidobacterium breve H4-2 and H9-3 in a mouse model of DSS-induced colitis, which is a representative animal model for ulcerative colitis. The author show that both strains significantly increase anti-inflammatory markers, and significantly decrease pro-inflammatory makers. For instance, DAI scores, colon length, weight loss, and histopathological analyses were performed as well as molecular analyses such as the presence of cytokines and expression (mRNA) of several markers (TLR’s and tight junction proteins). Finally, changes in the gut microbiota of the mice are investigated and correlated by a network analysis.  

Even though several studies have already described the anti-inflammatory effects of B. breve in the treatment of (DSS-induced) colitis [1-7], the present work hypothesizes that the beneficial effect of the strains might be related to the exopolysaccharide production of the two strains. Moreover, in nearly all analyses, the author compared the DSS- and B. breve-treated group with a mesalazine-treated group. Mesalazine is a gold standard therapy in the clinical practice of IBD. It is a topically active drug with no substantial systemic absorption [8], and therefore, it is a very relevant treatment arm to substantiate the local anti-inflammatory effects of the B. breve strains. Therefore, I think that the results of the present study are valuable and not only give insight into the mechanism of action of the B. breve strain, but also the mechanism of action of mesalazine.

However, I think a revision should take place based on the suggestions given below.

General points:

I think that the English in the article is sometimes a bit clumsy and not scientific. Please consider to assess your manuscript by a native English speaker. Just a few examples to show what I mean:

1.       Line 74: “cann’t”

2.       Line 257: “The melasazine, H4-2 and H9-3 intervention significantly improved weight loss (Fig 2A).” The intervention significantly reduced weight loss, and did not “improve” it.

3.       Line 265: “while intervention with H4-2 and H9-3 significantly increased the colon length of the mice (p < 0.05)”. The intervention did not result in an “increased” colon length of the mice, but reduced shortening of the colon.

4.       Line 322: “Miserably”. In my opinion this is not scientific writing.

5.       The entire conclusion. Please consider rewriting the conclusion and getting advice from a native English speaker. One example: “Among, H4-2 had better ameliorate effect.”

Throughout the manuscript, the use of “Bifidobacterium breve” is inconsistent. Sometimes it is written as “B. breve” and sometimes as “Bifidobacterium breve”. Consider writing “Bifidobacterium breve” once and thereafter only using “B. breve”.

Throughout the manuscript, the statistical significance of the experimental arms is unclear for me. The statistical significance is shown with lower case letters such as “a”, “b” etc. However, it is unclear what groups differ significant from each other. To give an example, see Figure 1C and the H9-3 box. What does “ab” mean? Significantly different from “c”? or “a” and “b”? Please consider clarifying the presentation of the statistical significance of the experiments throughout the entire manuscript.

The author both use “SCAF” and “SCFA” to describe “short chain fatty acids”. Only “SCFA” is correct. Please revise the manuscript accordingly and don’t use “SCAF”.

The authors hypothesize that EPS production might be correlated with the anti-inflammatory effects of the investigated B. breve strains. However, no EPS analysis in the colon of the mice are presented. Did the authors consider to perform these analyses or were these analyses performed? These data might not only present a dose-effect relationship, but would substantiate the hypothesis of the authors. If the authors have these data, please include it in the manuscript.

The author used the two strains H4-2 and H9-3, which produced 552.53 ± 12.54 and 363.17 ± 14.67 mg/L EPS. Nearly all results show that the anti-inflammatory effects of H4-2 are more prominent than the effects see with H9-3. This suggests that the there is a dose-effect relationship of EPS on the observed anti-inflammatory effects. However, I think that the authors are not highlighting this observation. It might be that the author have a different view than I have, but I am making this comment for the author to consider highlighting this observation throughout the manuscript.

Abstract:

Line 10 states “to explore the discrepancy”. I think that the word is used incorrectly.

Line 15 state “indicated that both B. breve strains had potential.” However, before line 15 there is no mention of the strains H4-2 and H9-3 and the use of “both” is therefore confusing.

Introduction:

Line 53 states “Among them, some probiotics have the same therapeutic effect as 5-aminosalicylic acid, and are recommended by the European Society as enteral and parenteral nutrition for the treatment of IBD.” However, there is no reference and what “European Society” do the authors mean? ECCO? Please insert reference and mention the correct society.

I think a paragraph should be dedicated to EPS: what are EPS, what is the putative mechanism of action, how are they produced by bacteria, and why do the authors want to investigate EPS-producing B. breve strains in the present study.

Material and methods:

Line 84 states “Bifidobacterium breve H4-2 and Bifidobacterium breve H9-3 with EPS production of 552.53 ± 12.54 and 363.17 ± 14.67 mg/L were isolated from the feces” I am curious how the author analyzed the EPS production? What EPS did they analyze with what method? How many samples were analyzed and averaged to get the presented numbers?

Some of the abbreviation are not clarified when they are first mentioned. For instance, line 167 mentions CAT, GSH, SOD and MDA without clarifying them. Please revise the manuscript accordingly: write out every abbreviation that you use for the first time, i.e. superoxide dismutase (SOD).

Results:

Line 240 states “this ratio was adopted for subsequent experiments”. What ratio? It is not mentioned whether 10:1, 100:1 or 1000:1 (Figure 1A) was used for further experiments.

Figure 2A: the authors state that  all treatments arms significantly reduced weight loss (line 255-258). However, I do not see statistical markers in the Figure 2A. Moreover, judging by the eye (which is admittedly not a scientifically sound method) all data points look similar. On what day(s) was the statistics performed to investigate the effect on weight loss?

Figure 8, 9, and 10 do not contain the mesalazine data. If the authors have these data, please present these data in the figures since this is valuable and interesting to show.

Discussion:

Lines 522-554 is a very long and incoherent paragraph. Please consider to rewrite this paragraphs and to structure it in multiple paragraphs instead of one.

Reference of this review report:

[1] 10.1038/s41598-020-75702-5

[2] 10.1007/s00394-020-02252-x

[3] 10.1038/pr.2015.115

[4] 10.1177/039463201402700418

[5] 10.1371/journal.pone.0095441

[6] 10.1371/journal.pone.0005184

[7] 10.1002/ibd.22848

[8] 10.1002/14651858.CD000543.pub5.

Author Response

Dear Reviewer:

Thank you for your comments concerning our manuscript entitled“Alleviation effects of Bifidobacterium breve on DSS-induced colitis depending on mucosal and epithelial barrier maintenance and gut microbiota modulation”(nutrients-1874332). Those comments are all valuable and very helpful for revising and improving our paper, as well as the important guiding significance to our researches. We have considered comments carefully and have made correction which can meet with approval. The main corrections in the paper and the responds to the comments are as following:

Responds to reviewer 2

I think that the English in the article is sometimes a bit clumsy and not scientific. Please consider to assess your manuscript by a native English speaker.

Reply: Thank you for your constructive suggestion.

We made our best effort to revise and correct paper writing. As you see, a large number of sentences in the revised manuscript were corrected. Due to too many revision have been done, we do not list detailed revision information here.

Thanks!

The reviewer’s comment: Throughout the manuscript, the use of “Bifidobacterium breve” is inconsistent. Sometimes it is written as “B. breve” and sometimes as “Bifidobacterium breve”. Consider writing “Bifidobacterium breve” once and thereafter only using “B. breve”.

Reply: Thanks for your kindly comment.

We have modified Bifidobacterium breve to B. breve in multiple place and highlighted in red.

Throughout the manuscript, the statistical significance of the experimental arms is unclear for me. The statistical significance is shown with lower case letters such as “a”, “b” etc. However, it is unclear what groups differ significant from each other. To give an example, see Figure 1C and the H9-3 box. What does “ab” mean? Significantly different from “c”? or “a” and “b”? Please consider clarifying the presentation of the statistical significance of the experiments throughout the entire manuscript.

Reply:  Thanks for your kindly comment.

In this study, SPSS 20.0 statistical software to perform Duncan’s multiple range test, among which, those marked with different lowercase letters such as a, b, c, d, e, etc. indicate significant differences (p < 0.05), and those marked with the same letters are such as a, ab, abc show that there is a difference, but the difference is not significant (p > 0.05).

As can be seen from Figure 1C, compared with LPS group, the intervention of H9-3 obviously reduced the content of IL-8, but the difference was not significant (p > 0.05), while the difference between H4-2 and LPS group was significant (p < 0.05). Sorry for the confusion, we have explained it in line 264-268.

Thanks!

The author both use “SCAF” and “SCFA” to describe “short chain fatty acids”. Only “SCFA” is correct. Please revise the manuscript accordingly and don’t use “SCAF”.

Reply:Thanks for your kindly comment.

We apologize for errors in manuscript, we have already modified all SCAF to SCFA and highlighted in red.

Thanks!

The authors hypothesize that EPS production might be correlated with the anti-inflammatory effects of the investigated B. breve strains. However, no EPS analysis in the colon of the mice are presented. Did the authors consider to perform these analyses or were these analyses performed? These data might not only present a dose-effect relationship, but would substantiate the hypothesis of the authors. If the authors have these data, please include it in the manuscript.

Reply: A good suggestion.

We would like to thank the reviewer for raising this critical suggestion. The role of EPS in intestinal barrier repair, the alleviation of colitis and analysis in the colon of the mice is under study, we prefer reporting these data in another manuscript and mainly focusing our object of this manuscript on the the different intestinal barrier repair mechanisms of strains with different EPS production performance on colitis.

Thanks again!

The author used the two strains H4-2 and H9-3, which produced 552.53 ± 12.54 and 363.17 ± 14.67 mg/L EPS. Nearly all results show that the anti-inflammatory effects of H4-2 are more prominent than the effects see with H9-3. This suggests that the there is a dose-effect relationship of EPS on the observed anti-inflammatory effects. However, I think that the authors are not highlighting this observation. It might be that the author have a different view than I have, but I am making this comment for the author to consider highlighting this observation throughout the manuscript.

Reply: Thanks for your kindly comment

We agree your comment very much, we have already highlight this view in line 598-600.

Thanks again!

Line 10 states “to explore the discrepancy”. I think that the word is used incorrectly.

Reply:Thanks for your kindly comment

We have modified it and highlighted in line 10.

Thanks!

Line 15 state “indicated that both B. breve strains had potential.” However, before line 15 there is no mention of the strains H4-2 and H9-3 and the use of “both” is therefore confusing.

Reply: Thanks for your kindly comment

We also apology for the sentences making the confusion about our study, we have already modified and highlighted in line 15.

Thanks!

Line 53 states “Among them, some probiotics have the same therapeutic effect as 5-aminosalicylic acid, and are recommended by the European Society as enteral and parenteral nutrition for the treatment of IBD.” However, there is no reference and what “European Society” do the authors mean? ECCO? Please insert reference and mention the correct society.

Reply:Thanks for your kindly comment

We apologize for errors in manuscript, we have changed, added the references and highlighted in red in line 56-57.

Thanks!

I think a paragraph should be dedicated to EPS: what are EPS, what is the putative mechanism of action, how are they produced by bacteria, and why do the authors want to investigate EPS-producing B. breve strains in the present study.

Reply:A good suggestion

We would like to thank the reviewer for raising this critical suggestion. A paragraph have already be dedicated according to your comment to explain what are EPS, how is EPS produced, and the possible underlying mechanism of EPS specifically in intestinal barrier repair.

The specific reason why we want to study EPS-producing Bifidobacterium breve is that in the process of screening EPS production, we found that two Bifidobacterium breve have great differences in EPS-producing ability, and at the same time, the fermentation characteristics of the two strains are also different. Therefore, we would like to explore the specific repair mechanism of the two strains for intestinal barrier damage, which is also explained in line 77-90 of the manuscript.

Thanks!

Line 84 states “Bifidobacterium breve H4-2 and Bifidobacterium breve H9-3 with EPS production of 552.53 ± 12.54 and 363.17 ± 14.67 mg/L were isolated from the feces” I am curious how the author analyzed the EPS production? What EPS did they analyze with what method? How many samples were analyzed and averaged to get the presented numbers?

Reply: A good suggestion

The exopolysaccharides were extracted by centrifugation, deproteinization, concentration, alcohol precipitation water solubility and dialysis, the data are expressed as the mean ± SD of 3 biological replicates. We also supplemented the reference about EPS extraction methods in line 98-100.

Thanks again!

Some of the abbreviation are not clarified when they are first mentioned. For instance, line 167 mentions CAT, GSH, SOD and MDA without clarifying them. Please revise the manuscript accordingly: write out every abbreviation that you use for the first time, i.e. superoxide dismutase (SOD).

Reply: Thanks for your kindly comment

We apologize for errors in manuscript, we also changed and highlighted in red in the line180-181.

Thanks again!

Line 240 states “this ratio was adopted for subsequent experiments”. What ratio? It is not mentioned whether 10:1, 100:1 or 1000:1 (Figure 1A) was used for further experiments.

Reply: Thanks for your kindly comment

We apology for the sentence making the confusion in our manuscript. This ratio is the ratio of bacteria to cells with the best IEC-6 proliferation effect screened above, and we have indicated it in line 256-258.

Thanks again!

Figure 2A: the authors state that all treatments arms significantly reduced weight loss (line 255-258). However, I do not see statistical markers in the Figure 2A. Moreover, judging by the eye (which is admittedly not a scientifically sound method) all data points look similar. On what day(s) was the statistics performed to investigate the effect on weight loss?

Reply: Thanks for your kindly comment

We also apology for the sentences and figure making the confusion about our study, we have already modified this sentence and explained in line 279-283 of the manuscript.

Thanks!

Figure 8, 9, and 10 do not contain the mesalazine data. If the authors have these data, please present these data in the figures since this is valuable and interesting to show.

Reply: Thanks for your kindly comment

We believe that mesalazine is a classic medicine for the treatment of colitis. Although bifidobacteria has a certain relieving effect, but it cannot be used as a medicine, meanwhile, it cannot possess therapeutical effect. Mesalazine only was served as a positive control in this study, so it was not necessary to analyze the level of short acid and the level of intestinal flora in the mesalazine group. Another reason is that mainly focusing our object of this manuscript on the repair mechanism of strains with different EPS production performance on colitis, therefore, the effect of mesalazine on short-chain fatty acids and intestinal microflora was not analyzed. A similar situation was also found in the following studies[1-3].

Thanks!

Lines 522-554 is a very long and incoherent paragraph. Please consider to rewrite this paragraphs and to structure it in multiple paragraphs instead of one.

Reply: Thanks for your kindly comment

We have made major revisions according to you comment in this section. Please see the revisions in the line 560-591 in the manuscript.

Thanks!

[1] Shi, J.; Xie, Q.; Yue, Y.; Chen, Q.; Zhao, L.; Evivie, S.E.; Li, B.; Huo, G. Gut microbiota modulation and anti-inflammatory properties of mixed lactobacilli in dextran sodium sulfate-induced colitis in mice. Food Funct. 2021, 12, 5130–5143.

[2] Chen, Y.; Yang, B.; Stanton, C.; Ross, R.P.; Zhao, J.; Zhang, H.; Chen, W. Bifidobacterium pseudocatenulatum Ameliorates DSS-Induced Colitis by Maintaining Intestinal Mechanical Barrier, Blocking Proinflammatory Cytokines, Inhibiting TLR4/NF-κB Signaling, and Altering Gut Microbiota. J. Agric. Food Chem. 2021, 69, 1496–1512.

[3] Chen, Y.; Jin, Y.; Stanton, C.; Ross, R.; Wang, Z.; Zhao, J.; Zhang, H.; Yang, B.; Chen, W. Dose-response efficacy and mechanisms of orally administered CLA-producing Bifidobacterium breve CCFM683 on DSS-induced colitis in mice. J. Funct. Foods. 2020, 75, 104245.

Round 2

Reviewer 1 Report

Authors have addressed my comments and suggestion